# MD Simulation Studies for Selective Phytochemicals as Potential Inhibitors against Major Biological Targets of Diabetic Nephropathy

**DOI:** 10.3390/molecules27154980

**Published:** 2022-08-05

**Authors:** Mohd Adnan Kausar, Sadaf Anwar, Wafa Ali Eltayb, Mohammed Kuddus, Fahmida Khatoon, Amr Ahmed El-Arabey, Amany Mohammed Khalifa, Moattar Raza Rizvi, Mohammad Zeeshan Najm, Lovnish Thakur, Subhabrata Kar, Mohnad Abdalla

**Affiliations:** 1Department of Biochemistry, College of Medicine, University of Hail, Hail 2440, Saudi Arabia; 2Biotechnology Department, Faculty of Science and Technology, Shendi University, Shendi 11111, Sudan; 3Department of Pharmacology and Toxicology, Faculty of Pharmacy, Al-Azhar University, Cairo 11751, Egypt; 4Department of Pathology, College of Medicine, University of Hail, Hail 2440, Saudi Arabia; 5Department of Physiotherapy, Faculty of Allied Health Sciences, Manav Rachna International Institute of Research & Studies, Faridabad 121004, India; 6School of Biosciences, Apeejay Stya University, Gurugram 122103, India; 7Pediatric Research Institute, Children’s Hospital Affiliated to Shandong University, Jinan 250022, China

**Keywords:** diabetic nephropathy, phytochemicals, molecular docking, molecular dynamics simulation, in-silico study

## Abstract

Diabetes is emerging as an epidemic and is becoming a public health concern worldwide. Diabetic nephropathy is one of the serious complications of diabetes, and about 40% of individuals with diabetes develop diabetic nephropathy. The consistent feature of diabetes and its associated nephropathy is hyperglycemia, and in some cases, hyperamylinemia. Currently, the treatment includes the use of medication for blood pressure control, sugar control, and cholesterol control, and in the later stage requires dialysis and kidney transplantation, making the management of this complication very difficult. Bioactive compounds, herbal medicines, and extracts are extensively used in the treatment and prevention of several diseases, and some are reported to be efficacious in diabetes too. Therefore, in this study, we tried to identify the therapeutic potential of phytochemicals used in in silico docking and molecular dynamic simulation studies using a library of 5284 phytochemicals against the two potential targets of type 2 diabetes-associated nephropathy. We identified two phytochemicals (i.e., gentisic acid and michelalbine) that target human amylin peptide and dipeptidyl peptidase-4, respectively, with good binding affinity. These phytochemicals can be further evaluated using in vitro and in vivo studies for their anti-hyperglycemia and anti-hyperamylinemia effects.

## 1. Introduction

In recent years, diabetes has become an epidemic and a public health concern worldwide. The estimated global incidence and prevalence of diabetes were estimated to be 26.6 million and 570.9 million, respectively, by 2025 [1]. Diabetes is considered to be responsible for 80% of death associated with premature non-communicable diseases (NCDs), including others such as cancer and cardiovascular and respiratory diseases [2]. There has been an increase in the global burden of diabetes, which will be soaring high in the future. Additionally, according to the WHO, by 2030, it will become the seventh-largest cause of death globally [3].

Diabetes mellitus (DM) is an endocrine disease caused by chronic hyperglycemia associated with a relative or absolute insulin deficiency, i.e., there can be a defect in insulin secretion, its action (insulin resistance), or both [4]. The glucose levels are regulated by the negative feedback mechanism, which regulates the levels of insulin and glucagon in the blood, thereby maintaining homeostasis. The disruption in this regulatory pathway results in altered glucose levels in the bloodstream, resulting in various complications [5]. The growing incidence of DM is a global health issue that appears to be caused by lifestyle changes, rising obesity rates, and population aging [1,6]. Diabetes is classified into two types, the first one being type I DM, which is an autoimmune disorder wherein the β cells are destroyed, affecting the secretion of insulin [7]. The second is type II DM, where the insulin secretion is reduced or resistance is developed against insulin, thereby hindering its action [8]. As a result, people with diabetes are at an increased risk of diabetic retinopathy, neuropathy, nephropathy, and cardiovascular diseases [9,10]. 

Osteopenia and bone fragility are some of the problems faced by many people with diabetes, which are caused either by residual insulin secretion or high insulin requirements. Apart from osteopenia, chronic hyperglycemia affects hemodynamic and metabolic homeostasis, which results in vascular dysfunction [11,12]. Diabetic nephropathy is the major cause of end-stage renal disease brought on by diabetes mellitus [13].

Diabetic nephropathy (DN) is characterized by changes in the kidney structure and is diagnosed by a group of clinical symptoms, primarily the persistent microalbuminuria that occurs in both, type I and type II DM. It involves increased levels of albuminuria, i.e., more than 300 mg/24 h in the urine [14]. The presence of albumin in the urine suggests that the globular filtration rate has decreased, which is another DN feature affecting renal function. The increased glucose levels activate various pathways such as protein kinase C and polyol and produce reactive oxygen species [15]. The elevated levels of cytokines and chemokines increase the vascular permeability, eventually leading to podocytopathy, thickening of the glomerular basement membrane, Kimmelstiel–Wilson (K-W) nodule formation, and glomerular mesangial matrix expansion [16,17]. One of the reasons for the increasing death rate among diabetes mellitus patients is primarily caused by diabetes nephropathy [18].

Over time, at least 40% of individuals with diabetes develop diabetic nephropathy [19]. There is a need to understand the link between diabetic nephropathy and diabetes mellitus to bridge the gap and develop effective therapeutics to cure the patient and improve their life [20,21]. However, controlling the glucose levels has been associated with halting the progression of diabetic nephropathy, although the exact underlying mechanism is unknown [22]. Another feature reported in the case of DN is the deposition of amylin amyloids in the kidneys [23].

Overall, two main features can be linked to cases of diabetes nephropathy condition: (1) hyperglycemia; (2) hyperamylinemia. Therefore, targeting these two factors can effectively control the progression of diabetic nephropathy. 

### Current Treatment Problems

Optimal cholesterol and blood pressure management, exercise, diet, and lifestyle therapies are advised to control diabetes-related complications such as diabetic nephropathy [24]. However, no therapeutic intervention is available for this diabetes nephropathy, and only symptomatic relief and hyperglycemic control medicines are provided. 

From ancient times naturally occurring bioactive compounds, i.e., secondary metabolites and plant extracts, have been used for their medicinal properties [25]. In the pharmaceutical industry, 30–50% of compounds are derived from herbal medicines and extracts [26]. Strategizing these photochemical against the two important targets of diabetes associated with nephropathy will be of great significance. A wide range of natural compounds identified in several plant species (*Carthamus tinctorius*, *Acacia Arabica*, *Aegle marmelose*, *Azadirachta indica*, and *Caesalpinia bonducella* etc.) have shown therapeutic value in the case of diabetes [27,28].

In the current research paper, we tried to evaluate the in silico activity of photochemicals against the major targets (DPP4 and amylin) of type II diabetes involved in diabetic nephropathy. Dipeptidyl-peptidase 4 (DPP4), a glycoprotein, acts as an exopeptidase that cleaves neuropeptides, incretin hormone, glucagon-like peptide, and glucose-dependent insulinotropic polypeptide (GIP) [29]. The release of GLP-1 and GIP peptides is important for maintaining the glucose level in the blood, as they control the insulin and glucagon secretion from the pancreas. The excess degradation of GLP-1 and GIP by DPP-4 can be related to the hyperglycemia condition, which can be corrected by blocking the DPP-4 activity. It is targeted by inhibitors such as the gliptin family of drugs for controlling type 2 diabetes [30]. Recent reports have shown the benefits of giving DPP4 inhibitors to diabetic nephropathy patients. In US, there has been a significant increase in the uptake of new diabetic medications targeting DPP4, which have been reported to be beneficial in improving patient outcomes with DN [31]. In another retrospective, an observational cohort study was performed to evaluate the reno-protective effects of DPP-4 inhibitors, urine albumin excretion, and eGFR, which were found to be significantly reduced after the administration of DPP-4 inhibitors in T2DM patients [32]. The role of DPP4 and its inhibitors is now also being studied to target fibrotic and inflammatory pathways in diabetic nephropathy conditions, making it an upcoming target [33]. The C-reactive protein (CRP) has been found to be pathogenic, and its expression is significantly correlated with dipeptidyl peptidase-4 (DPP4) diabetic nephropathy patients, although the reason is unknown; however, studies performed on mice with DPP4 inhibitors have effectively blocked this CRP-driven DN [34]. A study on DPP-4 inhibition with angiotensin II receptor blockers showed significantly reduced urinary albumin excretion and oxidative stress in diabetic eNOS knockout mice [35]. DPP4 inhibitors have also been evaluated for their role in inflammation associated with DN, and it has been reported that they can attenuate the inflammasome activation and the progression of DN in T2DM mice [36]. Additionally, research groups are also working on fluorescent probes such as GP-DCMNH2 for the early detection of DPP4 as a biomarker for diabetic nephropathy, and the early evidence has shown stronger fluorescence signals in the kidneys and blood of mice with diabetic nephropathy [37]. There are ample pieces of evidence suggesting the role of DPP4 as a potent target for DN and a good target for our virtual screening study.

The other target is amylin (IAPP), an amino acid peptide hormone co-secreted with insulin [38]. Its levels have been reported to be reduced in type II diabetes, and its amyloid deposits have been found in the pancreatic and kidney tissues of diabetes nephropathy patients [23]. The amylin deposition was related to the disease severity, as shown in a study done on 149 patients with biopsy-proven diabetic nephropathy [23]. It has been discussed that controlling the deposition of amylin can help in protecting renal function. MicroRNA (MicroRNA-375)-based studies have also shown islet amyloid deposition in the pancreas, as well as in other organs [39]. These facts suggest the important role played by native amylin in controlling the glycemic level, i.e., the amylin amyloid formation can be considered one of the potent targets for diabetic nephropathy. Therefore, targeting amyloid formation can be one of the strategies to control diabetes-associated nephropathy, as shown in Figure 1 [40]. 

Based on the literature and clinical evidence, these two potential target proteins, i.e., DPP4 and amylin, can be modified to control the hyperglycaemic and hyperamylinemia conditions associated with diabetes-associated nephropathy [41].

## 2. Methodology

### 2.1. Target Receptor Preparation

The 3D structures of our target receptors, human DPP-4 (PDB ID-2ONC) and human amylin interaction (2L86) were downloaded from the Research Collaborator for Structural Bioinformatics Protein Data Bank (RCSB PDB) [42]. The downloaded structures were subjected to protein preparation and refinement using the Maestro package in Schrodinger Suites 202. Using the Schrodinger workflow [43], the missing hydrogen atoms and missing side chains were added, and the bond orders were assigned. The OPLS3e force field was used for re-strained minimization for the final refinement of the structure. The receptor grid was generated around the SY1 ligand for human DPP-4 (PDB ID-2ONC) and around the full amylin peptide.

### 2.2. Ligand Preparation

The LigPrep tool was used to prepare the 3D structures of 5284 photochemicals downloaded from the PubChem database (https://pubchem.ncbi.nlm.nih.gov/, accessed on 3 June 2022). The structures of all ligands were imported into the project table in maestro and prepared for docking using the OPLS3e forcefield for the final energy minimization step.

### 2.3. Docking Protocol

The docking study was performed and the binding affinities of 5284 photochemicals against both the receptors of DPP-4 and amylin peptide were calculated. For docking, the high-throughput virtual screening mode was selected to screen the photochemical library in the Glide software.

### 2.4. In silico Pharmacokinetic Assessment of Investigated Compounds (ADMET)

The Qikprop 2.5 tools from Schrodinger software were used to calculate the ADME properties of all phytochemicals, as the predictions of physico-chemically significant descriptors and pharmacokinetically relevant properties form significant features for the development of drugs [22]. QikProp is a quick, accurate, and authentic tool that provides a range of parameters for evaluating drug-like properties and comparing them with known drugs [44]. It evaluates such properties based on one of the most important rules, i.e., Lipinski’s rule [45], for defining a drug’s molecular properties as significant in evaluating a drug’s pharmaco-kinetics. We also used the admet SAR online server for the prediction of properties such as BBB permeability, cytochrome p450 inhibition, and acute oral toxicity.

### 2.5. Molecular Dynamics Simulation

The Desmond MD simulation package was used to perform the simulation study of the best ligand-receptor complex [46] (human DPP-4 michelalbine and human amylin gentisic acid complex) to evaluate their safety. General workflows such as solvating the complex using a system builder and adding counter ions to neutralize the system were used. To minimize the system, the steepest descent steps were used. Additionally, the system was gradually heated from 0 to 310 K, and before the run the system was allowed to run using the thermostat setting method and pressure relaxation method for 5 ns each, respectively. Overall, a simulation of 100 ns was performed, and 5000 frame trajectories were generated every 10 ps.

## 3. Results and Discussion

### 3.1. Docking Study

Our study involving the screening of phytochemicals suggests a promising binding affinity against the potential targets of diabetic nephropathy. As mentioned earlier, amongst the 5284 phytochemicals that were screened against the human DPP-4 and human amylin, michelalbine (Docking score: −7.1) and gentisic acid (Docking score: −6.5) show good binding affinity to their respective targets (Figure 2 and Figure 3). Gentisic acid is a phenolic acid found in various natural plants and fruits such as *Citrus* spp., *Vitis vinifera*, *Hibiscus rosa-sinensis*, pears, and some mushrooms [47]. Michelalbine, an aporphine alkaloid, can be isolated from *Chelonanthus albus*, *Annona cherimola*, *Liriodendron tulipifera*, and *Artabotryshexapetalus* [48].

### 3.2. ADMET Analysis

The pharmaceutically relevant properties were predicted in silico using the QikProp tool. It is an advanced tool that helps in calculating the drug-like parameters of novel compounds and helps in filtering out compounds at an early stage of drug discovery [49]. In our study, both molecules with the best binding affinity showed drug-like properties (Table 1) and also followed the Lipinski rule of 5. The compound michelalbine is predicted to be a substrate for CYP2D6 and CYP3A4 enzymes involved in the biotransformation process and can inhibit the CYP2D6 enzyme, while gentisic acid is neither a substrate of the CYP enzymes nor an inhibitor. However, category III toxicity was predicted for both of these compounds, which requires further in vitro evaluation, as these are phytochemicals from commonly used edible plant species.

### 3.3. MMGBSA Analysis

The MMGBSA binding energies (Table 2) were obtained for both targets, i.e., amylin and DPP-4, with their complexes over a period of 100 ns. For DPP-4, the Coulomb energy (Coulom b) made a major contribution, followed by the van der Waals force, while in the case of the Human amylin gentisic acid complex, the major contributing force was the van der Waalsforce. 

### 3.4. RMSD Analysis

The stability of ligand-protein interactions was investigated using an MD simulation study over 100 ns (Figure 4), and the interactions were analyzed using RMSD and RMSF analyses. For the human amylin–gentisic acid complex, the protein RMSD was stable for most of the duration, and at around 20 ns a fluctuation of 1.6 A was observed, but the ligand was found to fluctuate throughout the duration of the simulation. In the case of the DPP-4 and michelalbine complex, the protein and ligand RMSD values were very stable throughout the 100 ns, showing a good interaction. For the gentisic acid–amylin complex, the RMSD was not very stable, and the ligands showed fluctuations.

### 3.5. Ligand Properties

The ligand parameters such as RMSD, radius of gyration, polar surface area, and molecular surface area were calculated and analyzed, as shown in Figure 5 [50]. Fo rmichelalbine interacting with DPP-4, significantly less fluctuation was seen throughout the simulation and no intramolecular hydrogen bond was detected. For the gentisic acid–amylin interaction, a high fluctuation up to 20 ns was observed for the RMSD, PSA, and rGyr values, followed by a stable equilibrium stage throughout the entire simulation duration.

### 3.6. Protein-Ligand Interaction Contacts

The protein-ligand contact histograms were plotted for stable interactions during the simulation (Figure 6). The four major types of interaction, including ionic interactions, hydrophobic interactions, water bridges, and hydrogen bonds, were used to analyze the protein-ligand contacts. In the case of DPP-4 and michelalbine, the major amino acids were involved in the interactions using the H-bonds (GLN_153, GLU_191, THR_129, TYR_128), hydrophobic interactions (TRP_124, TYR_195, VAL_252), and water bridges (GLN_123, ASP_192, TYR_211, ARG_253). However, the amylin interacted with gentisic acid with mainly hydrogen bonds (LYS_1, CYS_2, SER_19, ASN_21 THR_29, ASN_31, ASN_35), water bridges (ASN_3, ARG_11, SER_19, ASN_21, SER_28, THR_30, TYR_37), and very less ionic interactions (LYS_1, ASN_22, SER_29).

This study suggests the idea of employing traditionally used phytochemicals and medicinal plants for their activity against targets of diabetes nephropathy. Although some early studies targeting diabetes nephropathy have been reported, such as whenAKT1 and MAPK8 were targeted with myriocin using network biology and molecular docking approaches [51]. In another study, GeneCards, OMIM, TTD, DisGeNET, and DrugBank were used to identify the potential target and it was reported that the Yishen capsule interfered with the HIF-1α and JAK/STAT signaling pathways [52]. In another attempt, lisinopril drug analogs were created by replacing functional groups to improve inflammation control in the case of diabetic nephropathy [53]. Pathogen pathways such asmTOR and reduced autophagy have also been targeted using compounds such as santalin A [54]. However, our study has the advantage of using a strategically chosen target, keeping in mind the pathophysiology of diabetic nephropathy—hyperglycemia and hyperamylinemia. 

## 4. Conclusions

The use of natural products has a successful history of efficacious effects on various diseases, including diabetes. However, very few studies have been reported explaining the role of natural compounds in diabetes associated with nephropathy. As hyperglycemia and amyloid deposits are the principal factors responsible for the structural alterations at the renal level, glycemic control remains the main target for therapy in patients with the potential for the development of diabetic nephropathy. Our virtual screening study of the phytochemical library against the two main targets, i.e., DPP-4 and human amylin, led to the determination of compounds (michelalbine and gentisic acid, respectively) with the best theoretical affinity, as described in the study. Further, more target information on these compounds is needed for drug interventions, long-term drug design, in vitro and in vivo evaluations, and preparation in terms of the clinical trial applications so that they can be used for diabetes-associated nephropathy.

## Figures and Tables

**Figure 1 molecules-27-04980-f001:**
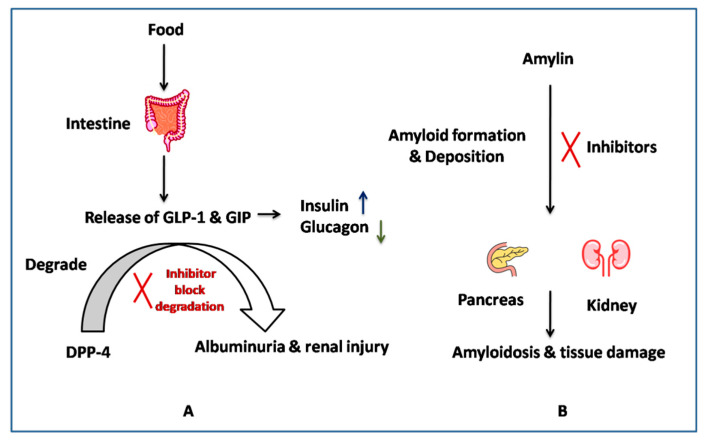
Action mechanism of (**A**) DPP-4, (**B**) Amylin inhibitors.

**Figure 2 molecules-27-04980-f002:**
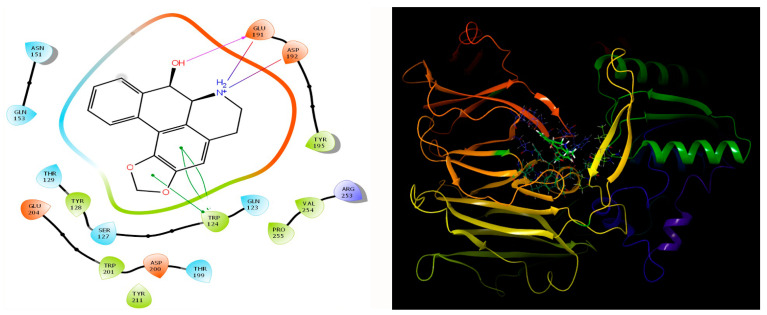
Interaction of human DPP-4 (PDB ID-2ONC) with michelalbine.

**Figure 3 molecules-27-04980-f003:**
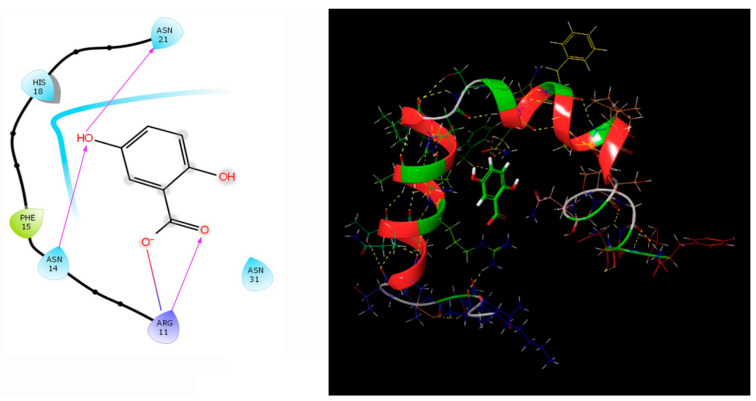
Human amylin’s interaction (2L86) with gentisic acid.

**Figure 4 molecules-27-04980-f004:**
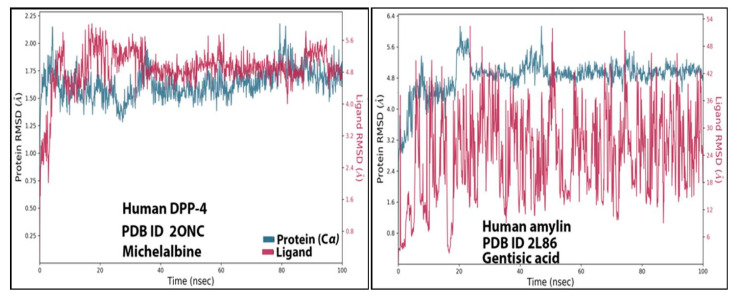
TheRMSD values of the complexes (human DPP-4 with michelabine and human amylin with gentisic acid).

**Figure 5 molecules-27-04980-f005:**
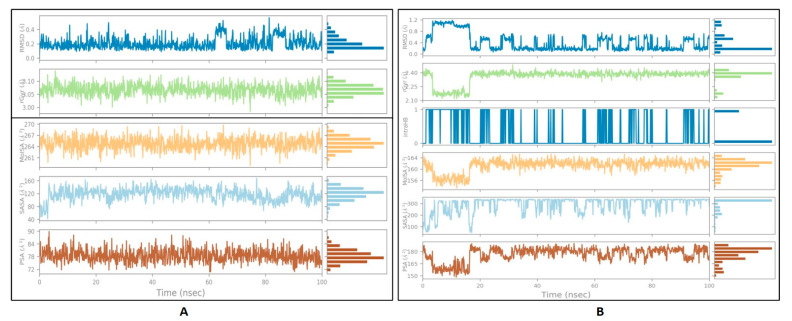
The properties of the ligands:(**A**) michelalbine; (**B**) gentisic acid.

**Figure 6 molecules-27-04980-f006:**
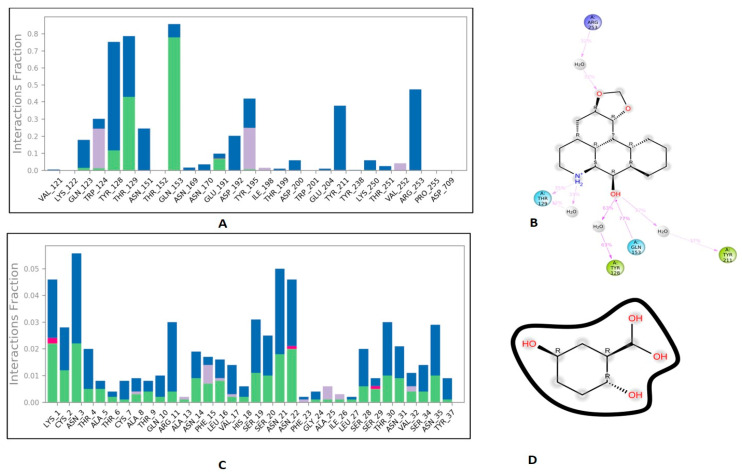
Important protein-ligand interaction residues:(**A**,**B**) PL contact for DPP-4–michelalbine complex; (**C**,**D**) PL contact for amylin–gentisic acid complex.

**Table 1 molecules-27-04980-t001:** ADMET results for the top molecules.

S.No	Receptor	Best Molecule	mol_MW	donorHB	accptHB	PSA	BBB	Cytochrome p450 Inhibition/Substrate	Oral Acute Toxicity
1	DPP-4	Michelalbine	281.31	2	4.7	52.139	0.9739	Substrate for CYP2D6 and CYP3A4/Only inhibit CYP2D6	Category-III
2	Amylin	Gentisic acid	154.122	2	2.5	91.598	0.9350	Non-inhibitor/non- substrate	Category-III

**Table 2 molecules-27-04980-t002:** Binding energies (MMGBSA) of the target and the best phytochemicals.

Target	Phyto-Chemical	MMGBSA dG Bind	MMGBSA dG Bind Coulomb	MMGBSA dG Bind Covalent	MMGBSA dG Bind Solv GB	MMGBSA dG Bind vdW
Human amylin	Gentisic Acid	0.057796044	0.015605996	0.030187095	0.043943182	−0.031940229
DPP-4	Michelalbine	−38.17881278	−50.49822686	0.999651095	54.78785013	−29.32973714

## Data Availability

The authors confirm that the data supporting the findings of this study are available within the article. Raw data that support the findings of this study are available from the corresponding author upon request.

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
