# Peer review of "MD Simulation Studies for Selective Phytochemicals as Potential Inhibitors against Major Biological Targets of Diabetic Nephropathy"

_molecules, 2022, doi:10.3390/molecules27154980_

Round 1

Reviewer 1 Report

In the manuscript “MD simulation studies for Selective Phytochemicals as Potential Inhibitors against Major Biological Targets of Diabetic Nephropathy” the authors have evaluated the therapeutic potential of phytochemicals using in silico docking and molecular dynamic simulation studies against the two potential targets of Type-2 Diabetes associated nephropathy. 

Although the data presented is of therapeutic interest, the manuscript must be improved before acceptance:

1. The authors have chosen two potential targets i.e. human amylin peptide and dipeptidyl peptidase-4 for Type-2 Diabetes associated nephropathy. The authors should further highlight the molecular mechanism explaining their role and provide more recent references regarding it.

2. The authors have discussed the predicted ADMET properties of the identified potential phytochemicals that are useful in filtering out compounds at early drug discovery pipeline. However, they have not provided information on other physiologically relevant properties such as BBB permeability, Cytochrome p450, Inhibition/Substrate, Oral Acute Toxicity. Computational data for these properties should be included and discussed further.

3. One of the important targets of therapeutic interest for controlling hyperglycemia is Sodium-glucose Cotransporter-2 (SGLT2), as it helps in renal tubular glucose reabsorption, thus increasing the blood glucose level. Inhibiting SGLT2 will be of great therapeutic significance, and the authors can consider it while targeting diabetic nephropathy.

4. Some minor grammatical mistakes have been observed in the text and the manuscript should be further revised to remove these mistakes in the final revised submission.

Reviewer 2 Report

In this study, the authors identified two phytochemicals from a library of 5284 phytochemicals through in silico docking and molecular dynamic simulation studies. These phytochemicals can be further investigated using in vitro and in vivo methods for the therapeutic potential for Type-2 Diabetes associated nephropathy. This study provided interesting results and the manuscript could be accepted after finishing the following minor corrections:

1.       Abstract should be short and precise, describing clearly the aim, major findings and conclusions of the research. The authors should not provide too much background information in the Abstract.

2.       Line 118, “1. Methodology” should be changed to “2. Methodology”; Line 127, the sentence should have a full stop at the end of it; Line 157, “1. Result & Discussion” should be changed to “3. Result & Discussion”

3.       The figure in the section of introduction should be removed.

Reviewer 3 Report

The manuscript by Mohd Adnan Kausar et al described in detail the MD simulation studies for Selective Phytochemicals as Potential Inhibitors against Major Biological Targets of Diabetic nephropathy

These findings could be very interested for future diabetic research scientist. The technical details are well presented, and the experimental analyses are described in depth.

  I believe that the manuscripts are written very well, supported with proper references / methodology and discussion. should be published 
